# Adjuvant Treatment with Tyrosine Kinase Inhibitors in Epidermal Growth Factor Receptor Mutated Non-Small-Cell Lung Carcinoma Patients, Past, Present and Future

**DOI:** 10.3390/cancers13164119

**Published:** 2021-08-16

**Authors:** Walid Shalata, Binil Mathew Jacob, Abed Agbarya

**Affiliations:** 1The Legacy Heritage Center & Dr. Larry Norton Institute, Soroka Medical Center, Beer Sheva 84105, Israel; 2Medical School for International Health, Ben-Gurion University of the Negev, Beer Sheva 84101, Israel; binil@post.bgu.ac.il; 3Bnai Zion Medical Centre, Oncology Division and Cancer Institute, Haifa 31048, Israel; abed.agbarya@b-zion.org.il

**Keywords:** lung cancer, adjuvant treatment, non-small-cell lung carcinoma (NSCLC), epidermal growth factor receptor (EGFR), tyrosine kinase inhibitor (TKI)

## Abstract

**Simple Summary:**

This review article details the progress of lung cancer treatments for a subtype known as non-small cell lung cancer with a special mutation of epidermal growth factor receptor (EGFR). In the following review article, we included the past trials that exclusively involved the chemotherapy treatments, present trials that explore the use of drugs known as tyrosine kinase inhibitors (different generations of EGFR- TKIs), as well as the ongoing trials that consider an interplay between these two. Finally, we propose some areas of future research given the implications of the current studies, namely regarding metastasis to the brain and central nervous system.

**Abstract:**

Lung cancer is the most common malignancy across the world. The new era in lung cancer treatments, especially this past decade, has yielded novel categories of targeted therapy for specific mutations and adjuvant therapy, both of which have led to improved survival rates. In the present study, we review the changes and development of treatments, with a special focus on adjuvant therapy using tyrosine kinase inhibitors (TKIs) administered to non-small-cell lung carcinoma patients who had a complete resection of the tumor harboring a mutated epidermal growth factor receptor. The clinical trials are dating from the past (chemotherapy trials), present (TKIs), and future (ongoing trials).

## 1. Introduction

Lung cancer is the leading cause of cancer-related deaths in the United States and poses a significant health care concern throughout the world [1]. Over 68% of patients are diagnosed after the age of 65 whereas less than 3% are diagnosed under the age of 45 years [2]. Non-small cell lung cancer (NSCLC) has the highest incidence of 85% among all lung cancers [3]. NSCLC includes any type of epithelial lung cancer apart from small-cell lung cancer and so is divided histologically into adenocarcinoma, squamous cell carcinoma (SCC), and large cell carcinoma [4]. As NSCLC is often insidious, patients can present with no symptoms until the disease is advanced, contributing to the poor prognosis of lung carcinoma [5]. Nearly 30% of patients with NSCLC have localized disease (stage I–IIIA) at the time of diagnosis and undergo curative surgery. Despite full tumor resection, many patients will experience systemic and/or local relapses, thereby succumbing to the disease. It is important to recognize that staging holds significance as to whether the tumor can be resected. Stages I and II are localized disease that can be resected without fear of the tumor having already metastasized whereas in stages IIIB and IV, resection is unfeasible. Stage IIIA is unique in that T3N0M0 is resectable whereas T3N2M0 is unresectable. Indeed, staging plays a role in determining the magnitude of the impact of the drug as a therapeutic. Several adjuvant therapies, including tyrosine kinase inhibitors (TKIs), chemotherapy, and immunotherapy, have been investigated as a means of improving survival outcomes for patients with fully resected NSCLC [6]. Currently, there is no consensus regarding optimal chemotherapy regimens for adjuvant treatment, especially when considering the added detail of specific tumor mutations. Clinical practice involves the combination of pharmaceutical agents such as cisplatin and second-generation chemotherapy drugs. Furthermore, the National Comprehensive Cancer Network guidelines lists options of chemotherapy regimens using cisplatin or carboplatin along with another drug, e.g., vinorelbine. Combinations of cisplatin with either etoposide (VP-16), gemcitabine, docetaxel, or pemetrexed (for adenocarcinoma) were also mentioned [7]. The most recent findings show that the combination of cisplatin plus vinorelbine is probably the best choice for adjuvant treatment [8,9].

Epidermal growth factor receptor (EGFR), also known as HER1, is a 170-kDa transmembrane receptor tyrosine kinase (RTK) found on the surface of epithelial cells and often overexpressed in malignancy [10]. Alterations in the EGFR gene have been found to be involved in cancer cell growth and tumor vascularization. EGFR mutations have been identified in up to 20% of all lung adenocarcinomas and there is a higher prevalence among females and non-smokers [11]. Patients with EGFR-mutant lung adenocarcinomas have a 70% response rate to first-line EGFR-TKI therapy, such as erlotinib, gefitinib, or afatinib [12]. TKI were previously used as supportive therapy for patients with NSCLC and, as shown in the chronological layout of the trials presented here, have since been used as monotherapy. There are five TKI currently available for the treatment of NSCLC in patients with EGFR-mutations, which are divided into three generations. The first generation includes erlotinib and gefitinib, the second generation dacomitinib and afatinib, and the third generation osimertinib. However, only osimertinib has been approved as adjuvant treatment for EGFR mutations in NSCLC [13]. Several clinical trials have shown improved efficacy, better outcomes in progression-free survival (PFS) and/or overall survival (OS) for several generations of TKI as compared to patients with EGFR-mutant lung adenocarcinoma who received chemotherapy as adjuvant treatment [14,15,16,17]. Additionally, these trials showed longer OS for patients with EGFR mutations compared to EGFR-wild type during the incidence of brain metastasis (patients were in stage I–III before brain metastasis) [18]. Given that there is propensity to acquired resistance, there is great need for continued advancements in therapeutic innovation, both in discovery and appropriate combinations.

NSCLC two-year survival rates have increased from 34% for diagnoses made in 2009–2010 to 42% for diagnoses made in 2015–2016 [3]. The new era of oncology treatments has included novel adjuvant therapy such as TKI in EGFR-mutant NSCLC. In this review paper we examine past, present and future therapies that have shown treatment efficacy for NSCLC patients with resected EGFR mutations such as those in exon 19 or 21 or the L858R substitution (Table 1).

In the current review, we have analyzed the data from clinical trials, uncharacterized for mutated epidermal growth factor receptor in the past, and highlighted the progress made in the treatment of non-small-cell lung carcinoma (NSCLC), with a special focus on the addition of adjuvant therapy using tyrosine kinase inhibitors (TKIs) with or without chemotherapy administered to NSCLC patients who had a complete resection of the tumor harboring mutated EGFR.

## 2. Past Clinical Trials That Included Adjuvant Chemotherapy (AC) Uncharacterized for EGFR Status in Patients with Completely Resected NSCLC

### 2.1. The Adjuvant Lung Project Italy Trial (ALPI)

The ALPI was an AC trial reporting participation of 1209 patients [19]. The prospective study included NSCLC cases diagnosed at stages I–IIIA, who were randomized to chemotherapy arm of mitomycin, vindesine, and cisplatin (MVP) for three cycles or no treatment. Resection involved lobectomy or pneumonectomy, although more limited types were allowed. Relevant lymph nodes were removed also. The 71 participating centers (Italian as well as five from the European Organization for Research and Treatment of Cancer Lung Cancer Cooperative Group) were permitted adjuvant radiotherapy to their discretion; for the MVP treatment arm, this was initiated 3–5 weeks after last cycle while for the control arm, this was initiated at 4–6 weeks after surgery. At 64.5 months of median follow-up, there were no differences in overall survival (hazard ratio (HR) 0.96, 95% confidence interval (CI), 0.81–1.13, *p* = 0.59) or PFS (HR 0.89, 95% CI, 0.76–1.03, *p* = 0.13) between the arms in this study (Table 2). Hematologic toxicity was the most frequently observed adverse effect, while other adverse effects included grade 2 and 3 acute pneumonitis, grade 4 pneumonitis and grade 4 esophagitis. Both treatment and control arms showed >40% of patients with brain relapse (Table 3).

### 2.2. International Adjuvant Lung Cancer Trial (IALT)

The IALT has provided the most evidence for AC in this setting to date [20]. There were 1867 patients with completely resected NSCLC classified as stages I–IIIA in this prospective study that compiled data from multiple participating centers. Each center had license to determine the dose of cisplatin, the drug combined alongside and policy for postoperative radiotherapy. Patients were randomly assigned to the chemotherapy arm, with cisplatin being administered to 74% of them. Three or four courses of cisplatin-based chemotherapy were used, along with either etoposide, vinorelbine, vinblastine, or vindesine. The results showed that the AC arms had a higher survival rate than the observation arm (44.5% vs. 40.4% at five years, HR for death 0.86 (95% CI, 0.76–0.98, *p* < 0.03)) and exhibited better PFS (39.4 vs. 34.3 at five years HR 0.83 (95% CI, 0.74–0.94, *p* < 0.003)). This trial was successful in demonstrating better survival by cisplatin-based adjuvant treatment (Table 2). Moreover, 22.6% of patients presented with at least one instance of grade 4 toxic effect, including neutropenia, thrombocytopenia, and vomiting as the significant effects, which were consistently found in over 1% of patients (Table 3).

### 2.3. The Big Lung Trial (BLT)

The Big Lung Trial was a negative trial that failed to observe survival benefits for patients with stage I to III NSCLC with resection [21]. Only 381 patients were selected, so presumably the study did not meet quality standards especially given the various stratifications in therapeutic delivery. Preoperative chemotherapy was given to 3% of the patients, and 5% from both treatment arms did not achieve complete resections. Multiple cisplatin-based chemotherapy schemes were devised (e.g., cisplatin + vindesine, mitomycin + ifosfamide + cisplatin, mitomycin + vinblastine + cisplatin or vinorelbine + cisplatin) for use as the treatment arm, which was compared to the control arm that did not receive chemotherapy. Further, the patients were staggered into different categories of treatment schedules where 13% of patients did not receive scheduled chemotherapy, instead receiving radiotherapy, 21% receiving either one or two cycles of chemotherapy and 64% receiving all three cycles of the assorted therapy schemes. There was no benefit for chemotherapy protocol in terms of survival (HR 1.02; 95% CI, 0.77–1.35, *p* = 0.90) (Table 2). Moreover, 30% developed grades 3 and 4 toxicity, mainly hematologic, nausea/vomiting, and neutropenic fever. Additionally, six patients were reported dead by treatment-related toxicity (Table 3).

### 2.4. Cancer and Leukemia Group B (CALGB) 9633 Trial

This study recruited 344 stage IB patients with resected NSCLC [22]. Participants were randomized and assigned to two arms. The first arm received four cycles of paclitaxel with carboplatin chemotherapy while the second arm served as control under observation. Moreover, 57% received four cycles at full dose while 86% received all four cycles. Survival was not significantly different HR 0.83 (95% CI, 0.64–1.08, *p* = 0.125). Median survival times were 95 and 78 months in AC arm vs. observation. DFS was without statistically significant difference (HR 0.80; 90% CI, 0.62–1.02, *p* = 0.065). Median DFS was 89 and 56 months in AC and observation arm. However, for patients who had tumors ≥ 4 cm in diameter, there was a significant survival difference for the doublet chemotherapy agents (HR 0.69; 95% CI, 0.48–0.99, *p* = 0.043). OS analysis showed 31% lowered risk of death. Median survival time was 99 months in AC vs. 77 months in control arm. DFS was improved significantly by 31% in AC (HR, 0.69; 90% CI, 0.49–0.97; *p* = 0.035). Median DFS was 96 months in AC vs. 63 months in control group for those patients with tumors ≥ 4 cm, however there was no difference in DFS for patients with smaller tumors (Table 2). Moreover, 35% of patients developed grades 3 and 4 toxicity, predominantly neutropenia, and no treatment-related toxic deaths were recorded (Table 3).

### 2.5. JBR-10 Trial

In this study, 482 patients diagnosed with NSCLC who had complete tumor resection, were recruited [23]. Randomization arms were observation or AC. Four cycles of vinorelbine with cisplatin chemotherapy were administered. Moreover, 45% of the patients were at stage IB and 55% were at stage II (T3N0 patients were excluded). AC extended OS in treatment arm vs. control arm 94 vs. 73 months, (HR for death, 0.69 *p* = 0.04), significantly increased PFS, [HR for recurrence, 0.60 (95% CI, 0.45–0.79, *p* < 0.001)]. Survival at five years was for AC 69% (95% CI 62–75) vs. 54% (95%CI 48–61) (*p* = 0.03) as compared to observation. PFS at five years was 61% for AC (95%CI 54–68%) vs. 49% in the observation arm (95%CI 42–55%) *p* = 0.08. The subgroup analysis showed no survival benefit for patients with stage IB (*p* = 0.79). Nevertheless, for patients with stage II the chemotherapy arm showed significant benefit with a median survival of 80 months compared to 41 months for the observation arm (HR 0.59 (95% CI, 0.42–0.85, *p* = 0.004)) (Table 2). Grade 3 or 4 neutropenia was the most common adverse effect with 73% of patients developing it from chemotherapy. Other notable grade 3 or 4 adverse effects included anemia (7% of patients), fatigue (15%), nausea (10%), anorexia (10%), vomiting (7%), constipation (3%), sensory neuropathy 2%, motor neurotoxicity 3%, hearing loss 2%, febrile neutropenia following colony-stimulating factor administration (7%), and thrombocytopenia (1%). There were two toxicity related deaths (Table 3).

### 2.6. Adjuvant Navelbine International Trialist Association (ANITA) Trial

Briefly, 840 patients who underwent complete NSCLC resection were recruited to this study [24]. Patients with stage I to IIIA were randomized to a combination of cisplatin plus vinorelbine for four courses vs. observation. Moreover, 36% of patients had stage IB, 24% stage II and 39% had stage IIIA disease. The 101 participating centers across 14 countries were given the prerogative to determine whether postoperative radiation therapy (PORT) should be used: 24% of patients in the AC arm received PORT compared to 33% in the control arm. Interestingly, compliance to treatment was higher for cisplatin compared to vinorelbine. The results showed that the chemotherapy arm significantly improved median survival of 65.7 (95% CI 47.9–88.5) vs. 43.7 months. OS of an 8.6% improvement at five years (HR 0.80 (95% CI, 0.66–0.96, *p* = 0.02)) and 8.4% at seven years (Table 2). Moreover, 92% patients developed neutropenia, while febrile neutropenia was found in 7%, and seven toxicity related deaths occurred (Table 3).

### 2.7. MAGRIT Trial

In the MAGRIT trial, 12,820 patients with resected MAGE-A3 expressing NSCLC were recruited [30]. Of these, 33% presented with exclusively MAGE-A3 positive tumor. Patients, from 443 centers across 43 countries, with stage I to IIIA were either given adjuvant chemotherapy or not before being randomized and assigned to two arms. The first arm (AC) received 13 intramuscular injections of recMAGE-A3 with AS15 immunostimulant (MAGE-A3 immunotherapeutic) vs. the second arm of placebo for 27 months. In patients with MAGE-A3-positive surgically resected NSCLC, adjuvant treatment with the MAGE-A3 immunotherapeutic did not improve disease-free survival when compared to placebo, which showed 60.5 months (95% CI 57.2-not reached) for the MAGE-A3 arm and 57.9 months (55.7-not reached) for the placebo group (HR 1.02, 95% CI 0.89–1.18; *p* = 0.74). The use of the MAGE-A3 immunotherapeutic for use in NSCLC has stopped because of the findings showing no differences between MAGE-A3 immunotherapeutic in comparison to placebo (Table 2). Both arms expressed similar frequency of grade 3 or worse adverse events, such as infection/infestation, vascular disorders, and neoplasm (Table 3).

Reported adverse events (AE) of past clinical trials that included adjuvant therapy uncharacterized for EGFR status in patients with completely resected NSCLC are listed in Table 3.

## 3. EGFR-TKI Used in Clinical Trials for NSCLC Adjuvant Therapy

### 3.1. Erlotinib

Erlotinib is a derivative of quinazoline classified as an antineoplastic agent. It is a first generation TKI medication treatment for NSCLC tumors with EGFR mutations, exon 19 deletion (ex19del), or exon 21 point mutation (L858R). Erlotinib exerts its antagonist ability by competing with adenosine triphosphate (ATP) on the catalytic site of the EGFR located at the intracellular part [31]. Through reversible binding, erlotinib inhibits the phosphorylation of EGFR, thus disabling the signal transduction pathway and blocking proliferative cellular reactions leading to reduced carcinogenesis process related to activation of EGFR. This targeted therapy drug is orally administered. The Food & Drug Administration (FDA) approved erlotinib for NSCLC on 18 November 2004 and has, since 18 October 2016, restricted its use in lung cancer as a first line treatment to metastatic NSCLC with the EGFR mutations listed above. It remains first-line treatment for locally advanced, unresectable, or metastatic pancreatic cancer when combined with gemcitabine [32].

### 3.2. Gefitinib

Gefitinib is an anilinoquinazoline compound possessing antineoplastic properties. This drug belongs to first generation therapy of TKI NSCLC harboring EGFR exon 19 or exon 21 (L858R) mutation. Gefitinib specifically inhibits the catalytic activity of several tyrosine kinases among them EGFR. It is considered as an antagonist of EGFR and could cause an inhibition of tyrosine kinase-dependent tumor growth [25]. The drug is able to bind in a competitive way to the ATP domain of the tyrosine kinase part on EGFR, hence blocking the autophosphorylation of the receptor, and as a consequence inhibiting the signal transduction downstream cellular mechanism. Gefitinib actions include the induction of cell cycle arrest and restricting angiogenesis. It is given through oral administration. The FDA approved gefitinib for advanced NSCLC progressing beyond platinum doublet chemotherapy and docetaxel on 5 May 2003 and, since 13 July 2015, it has been expanded for use as a first line treatment in metastatic NSCLC with the EGFR mutations listed above [26].

### 3.3. Osimertinib

Osimertinib is a third generation EGFR inhibitor which can selectively bind in an irreversible way. It is indicated for patients suffering NSCLC as an antagonist agent with antitumoral capability. This TKI binds covalently to mutated EGFR in exon 19, exon 21 L858R, as well as to exon 20 T790M. Therefore, it may prevent cell signaling cascade mediated by EGFR activation [27]. Osimertinib potentially may inhibit neoplasm growth in EGFR-overexpressing tumor cells and induce cell death. This medication is orally available. Approval for osimertinib by the FDA came first on 13 November 2015 for use as adjuvant therapy after tumor resection in adult patients with NSCLC with the above EGFR mutations, but this was expanded on 18 April 2018 for use as a first line therapy in metastatic NSCLC with the EGFR mutations listed above [28].

## 4. Past Clinical Trials That Included Adjuvant Therapy Using EGFR-TKI in Patients with Completely Resected EGFR Mutated NSCLC

### 4.1. Pemetrexed-Carboplatin Adjuvant Chemotherapy with/without Gefitinib Trial

In this phase II study, 60 patients with resected NSCLC bearing EGFR mutations, exon 19 deletion or L858R, were enrolled [29]. Participants with stage IIIA were randomized to a combination of pemetrexed and carboplatin, for four cycles, followed with or without gefitinib for six months. The results showed longer PFS among those who received chemotherapy + gefitinib (median, 39.8 months) than among those who received only chemotherapy (27.0 months)(HR, 0.369; 95% CI, 0.161–0.847; *p* = 0.014). Two-year DFS rate was 78.9% in the AC treatment group with TKI vs. 54.2% without TKI. Two-year OS was 92.4% in the AC treatment arm with TKI vs. 77.4% in control arm without TKI (HR, 0.37; 95% CI 0.12–1.11, *p* = 0.076). OS was also longer for chemotherapy + gefitinib arm (median, 41.6 months) than chemotherapy alone (32.6 months, *p* = 0.066) (Table 4).

### 4.2. EVAN

Briefly, 102 patients with resected NSCLC harboring EGFR exon 19 deletion or L858R mutation were recruited [33]. Patients diagnosed at stage IIIA were randomized to a combination of vinorelbine and cisplatin, for four cycles vs. erlotinib (until disease progression). The results showed two-year DFS of 81.4% (95% CI 69.6–93.1) in the erlotinib arm and 44.6% (95% CI 26.9–62.4) in the chemotherapy arm (relative risk 1.823 95% CI 1.194–2.784; *p* = 0.0054) (Table 4).

### 4.3. ADJUVANT Trial (CTONG 1104)

This phase III study enrolled 222 patients with EGFR confirmed mutations (exon 19 deletion or L858R) in resected NSCLC [16]. Patients with stage II to IIIA were randomized to receive gefitinib or a combination of vinorelbine and cisplatin, for four cycles vs. gefitinib for two years. The results showed prolonged two-year median DFS of 28.7 months with TKI (95% CI, 24.94–32.46) vs. 18 months with chemotherapy combination (95% CI, 13.59–22.34), HR,0.60, (95% CI, 0.42–0.87, *p* = 0.005). No significant differences were found for the analysis of OS final results [16] (Table 4).

### 4.4. SELECT Trial

Briefly, 100 patients with resected NSCLC bearing mutant EGFR, were recruited to this study [17]. Participants diagnosed with stage IB to IIIA, after AC with or without radiotherapy, were randomized to a single arm of erlotinib for up to two years. The results showed two-year course was achieved in 69% of patients. DFS at two years was 88%. Patients’ median follow-up was 5.2 years. At five years, DFS was 56% (95% CI, 45–66%) and OS was 86% (95% CI, 77–92%). Recurrence of the disease was found in four patients while receiving therapy with erlotinib, and in 36 patients who concluded erlotinib treatment, having 25 months as median time to recurrence. Retreatment with erlotinib in 65% of the recurrent patients had a 13-month median duration (Table 4).

### 4.5. ADAURA Trial

In this phase III study, 682 patients with resected NSCLC, carrying EGFR-mutation (Ex19del or L858R) were recruited [14]. Participants with stage IB to IIIA after AC were randomized to osimertinib vs. placebo for three years. Results showed 89% of patients were disease-free (95% CI, 85–92) in the osimertinib arm and 52% (95% CI, 46–58) in the placebo arm at two years. Overall HR for disease recurrence or death of 0.20 (99.12% CI, 0.14–0.30; *p* < 0.001) can be translated into 80% lower risk for disease recurrence or death, thereby extending DFS in osimertinib arm vs. placebo. Furthermore, results were significant in showing that, at 24 months, 98% of the patients were alive without central nervous system (CNS) disease after receiving osimertinib vs. 85% of patients who received placebo. Overall HR for CNS disease recurrence or death, 0.18; (95% CI, 0.10–0.33) means that 82% decreased risk of CNS disease recurrence or death in the osimertinib arm (Table 4).

Adverse events reported in the clinical trials involving adjuvant therapy using EGFR-TKI in patients with completely resected NSCLC harboring EGFR-mutations are summarized in Table 5.

## 5. Ongoing Clinical Trials That Include Adjuvant Therapy Using EGFR-TKI in Patients with Completely Resected EGFR Mutated NSCLC

### 5.1. Adjuvant Lung Cancer Enrichment Marker Identification and Sequencing Trial (ALCHEMIST Trial)

In this phase III study, 450 patients with resected NSCLC are estimated to be enrolled [34]. Inclusion criteria are diagnosis of stage IB to IIIA with confirmed EGFR exon 19 or L858R mutations. Participants are randomized to two pairs of blinded and unblinded arms to erlotinib vs. placebo for up to two years. The primary objective of the trial is to examine if adjuvant therapy with erlotinib has an improved OS while secondary objectives consider better DFS, the safety profile of erlotinib, and the use of circulating EGFR mutations in cell-free plasma DNA as a prognostic marker. After treatment, patients will be followed up every six months for four years and then once a year for the next six years. Study outcomes are anticipated to be released on October 2026 (Table 6).

### 5.2. EMERGING Trial

The EMERGING Trial recruited 72 patients with resected NSCLC to this investigation [29]. Participants with stage IIIA-N2 NSCLC bearing EGFR mutation in exon 19 or 21 were randomized to erlotinib vs. combination of gemcitabine plus cisplatin. Erlotinib is first given as neoadjuvant for 42 days followed by one year as adjuvant therapy. Gemcitabine and cisplatin are given for two cycles of neoadjuvant therapy followed by a further two cycles of adjuvant therapy. The outcomes of the research include PFS and OS at three years. Post-surgery care for up to two years comprised of chest computerized tomography (CT) scan, abdominal ultrasound every three months, brain MRI bi-annually and bone scan once a year. Study results are expected to be published on December 2022 (Table 6).

### 5.3. Adjuvant Afatinib Trial

Briefly, 95 patients with resected NSCLC were enrolled [35]. Inclusion criteria were diagnosis of stage I to III NSCLC harboring EGFR mutations. Participants were randomized to short course (three months) afatinib vs. long course (two years) afatinib. Patients were followed up every six months for three years and then once in the fourth year. Chest CT scan, blood tests, performance status, and a physical exam were conducted at these follow ups. Study results are estimated for November 2021 (Table 6).

## 6. Discussion

Non-small cell lung cancer accounts for the majority of all lung cancers, which represent the leading cause of cancer-related deaths in the United States [3]. Traditionally, the disease has poor prognosis due to late onset of symptoms and curative resection is often complicated by systemic or local relapse [5]. Adjuvant therapies, including TKIs, chemotherapy, and immunotherapy, have been shown to improve survival outcomes. New research continues to increase the number of options for treatment. A particularly promising avenue of study involves the mutations of epidermal growth factor receptor in NSCLC as they can be better targeted by the generations of TKIs. Presently, drugs of the first TKI generation, erlotinib and gefitinib, along with the third-generation agent, osimertinib, have been approved for clinical use as adjuvant treatment due to their better efficacy, improved outcomes in PFS and/or OS [14,15,16,17]. These drugs are often indicated for specific *EGFR* mutations that activate the tyrosine kinase receptor (anywhere from autophosphorylation of the receptor domains to the recruitment and activation of proteins for downstream signaling) including those in exon 19 or 21 or the L858R substitution as clinical data have produced compelling evidence for their efficacy as monotherapy. The importance of molecular targeting (identifying presence of mutations) cannot be understated as there is evidence for mutual exclusiveness between *EGFR*-activating mutations and *KRAS* mutations that are often seen in adenocarcinoma. Ignoring molecular testing threatens to increase drug resistance to TKI.

Past studies involving adjuvant chemotherapy uncharacterized for EGFR mutations have had mixed results. They played an innovative role to provide clinical data and document the efficacy of different treatments including cisplatin, carboplatin, paclitaxel, vindesine, vinorelbine, vinoblastine, etoposide, ifosfamide, and mitomycin. While the ALPI, BLT, and later MAGRIT (involving immunotherapy subsequent to AC) were negative trials, the IALT, CALGB 9633, JBR-10, and ANITA trials demonstrated improvements in progression-free survival, disease-free survival, and overall survival. Other trials not included in this review have suggested the efficacy of additional chemotherapeutics, such as uracil/tegafur though these therapies have as yet not been approved for commercial use in North America or Europe [36]. The survival benefits accrued by AC are dependent on staging and tumor size, however the lack of attunement to gene markers and mutations leaves the therapeutic strategy lagging in the modern era of personalized medicine.

In the setting of non-EGFR mutant resected NSCLC and subsequent AC, one notable ongoing study has been IMpower010 which is similarly examining the efficacy and safety of drug, in this case the anti-PD-L1 monoclonal antibody atezolizumab, compared to best supportive care (BSC) following AC in resected NSCLC [37]. IMpower010 is the first phase III global study using an immune checkpoint inhibitor to show statistically significant DFS benefit vs. BSC for patients with stage IB to IIIA. Note that immune checkpoint inhibitors block the cell cycle to ensure greater regulation and/or destruction of aberrant cells whereas TKI specifically targets the EGFR-ErbB family of protein receptors within the cell for signal transduction and transcription factor recruitment. Comparable to the pemetrexed-carboplatin-gefitinib, EVAN, and ADJUVANT—CTONG 1104 trials, there has been increasing interest in identifying whether the use of TKI or other non-chemotherapeutic can produce improved survival rates with lower adverse events across all forms of NSCLC. All three of these studies have shown improvements in DFS while adverse events were comparable between TKI plus AC and AC alone. Notably the ADJUVANT—CTONG 1104 trial showed a decrease in serious adverse effects (7% vs. 23%). This will continue to be an area of investigation both for the use of TKI in EGFR-mutant NSCLC as well as non-EGFR mutant NSCLC.

Osimertinib, a third-generation adjuvant TKI therapy plays an innovative role as novel treatment strategy [14]. Osimertinib trial findings in ADAURA demonstrated increased efficacy such as significantly longer DFS in patients administered with this oral TKI in comparison to patients who received placebo. Previous studies have successfully shown long disease-free survival statistics for patients taking adjuvant erlotinib and gefitinib for a range of disease staging from IA to IIIA [16,17]. However, these patients are still susceptible to relapses that target CNS metastasis. The role of EGFR mutations, including but not limited to exon 19 deletion, as a predictor for such brain metastasis may be considered [18]. Osimertinib’s irreversible binding to EGFR, and potentially antineoplastic effects in curtailing tumor cell proliferation and tumor vascularization, should be further explored.

The National Institute of Health’s National Library of Medicine displays three ongoing studies involving erlotinib and afatinib which will yield better insight into the utility of these TKI drugs as adjuvant therapy in prolonging lifespan in patients who have been surgically resected for EGFR-mutant NSCLC across stages IA-IIIA [34,35,38]. These investigations will detail greater knowledge concerning the safety profile of erlotinib and examine the question pertaining to afatinib approval across both short and long course treatments. There is a scope for future studies assessing the efficacy and survival outcomes in dacomitinib, comparisons between TKI and cisplatin + vinorelbine as an adjuvant therapy, and further elucidation of brain and other metastasis site incidence.

## 7. Conclusions

In this review paper, we analyzed and summarized recent findings regarding the treatment and efficacy of TKI for NSCLC patients with EGFR mutations. Innovative pharmaceutical research of the 21st century has led to the development of modern drugs aimed at specific mutated proteins expressed in malignant tumors. These anticancer medications have benefited the clinical oncology field in prescribing a TKI adjuvant therapy regimen to patients with resected EGFR-mutated NSCLC. The advanced generation agents comply with a targeted treatment, patient-tailored approach, acting to extend life through higher efficacy, longer progression-free survival, and improved overall survival of NSCLC patients. Prospective investigational trials are imperative in efforts to overcome the challenges posed by third-generation EGFR-TKI resistance and the discovery of new mutations.

## Figures and Tables

**Table 1 cancers-13-04119-t001:** Reviewed clinical trials of adjuvant therapy for past trials that were uncharacterized for EGFR, past and present trials involving TKI with or without chemotherapy in completely resected EGFR-mutated NSCLC.

Treatment	Chemotherapy Uncharacterized for EGFR-TKI (*n* = 7)	TKI with or without Chemotherapy (*n* = 5)	TKI ^a^ (*n* = 3)
Trial Name ^b^	ALPI [18] IALT [19] BLT [20] CALGB 9633 [21] JBR-10 [22] ANITA [23] MAGRIT [24]	Pemtrexed + carboplatin + gefitinib [25] EVAN [26] ADJUVANT [15] SELECT [16] ADAURA [14]	ALCHEMIST [27] EMERGING [28] Afatinib [29]

^a^ Present clinical trials include ongoing and studies pending future results. ^b^ Superscript number. Abbreviations: EGFR, epidermal growth factor receptor; NSCLC, non-small-cell lung cancer; TKI, tyrosine kinase inhibitor.

**Table 2 cancers-13-04119-t002:** Past clinical trials that included adjuvant therapy uncharacterized for EGFR status in patients with completely resected NSCLC.

Outcome	Control Arm	Treatment Arm	NSCLC Stage	Participants (*n*)	Date Published	Trial Name
No differences found	No treatment	Mitomycin, vindesine and cisplatin	I to IIIA	1209	2003 October	The Adjuvant Lung Project (ALPI)
PFS is 39.4 vs. 34.3 at 5 years HR 0.83 (95% CI, 0.74–0.94, *p* < 0.003)]	Observation	Cisplatin-based with either etoposide, vinorelbine, vinblastine, or vindesine	I to IIIA	1867	2004 January	International Adjuvant Lung Cancer Trial (IALT)
No survival benefit found	No treatment	Cisplatin-based: Cisplatin + vindesinem Mitomycin + ifosfamide + cisplatin, Mitomycin + vinblastine + cisplatin, Or vinorelbine + cisplatin	I to IIIA	381	2004 July	The Big Lung Trial (BLT)
AC arm shows significantly improved survival of patients with tumors ≥ 4 cm	Observation	Paclitaxel+ Carboplatin	IB	344	2008 September	CALGB 9633 trial
Chemotherapy significantly prolonged OS, PFS	Observation	Vinorelbine + Cisplatin	IB to II	482	2005 June	JBR-10 trial
Chemotherapy significantly improved median survival	Observation	Vinorelbine + Cisplatin	I to IIIA	840	2006 August	Adjuvant Navelbine International Trialist Association (ANITA)
1. Found no differences between arms 2. Found no differences between arms in survival	Placebo	1. AC followed by MAGE-A3 2. No AC followed by MAGE-A3	IB to IIIA	12,820	2016 June	MAGRIT trial

Abbreviations: NSCLC, non-small-cell lung cancer; AC, adjuvant chemotherapy; PFS, progression-free survival; CALGB, Cancer and Leukemia Group B; OS, overall survival.

**Table 3 cancers-13-04119-t003:** Adverse events (AE) reported in past clinical trials that included adjuvant therapy uncharacterized for EGFR status in patients with completely resected NSCLC.

Trial	Chemotherapy Arm	Control Arm
The Adjuvant Lung Project (ALPI)	Grade 3 neutropenia in 16% of patients Grade 4 neutropenia in 12% of patients >40% of patients had relapse of cancer in the brain	>40% of patients had relapse of cancer in the brain
International Adjuvant Lung Cancer Trial (IALT)	Grade 4 toxicity in 23% of patients Approximately 1% deaths induced by toxic effects	NR
The Big Lung Trial (BLT)	Grade 3 and 4 toxicities in approximately 30% of the patients	NR
CALGB 9633 trial	Grade 3–4 neutropenia in 35% of patients. 34% required dose reduction	NR
JBR-10 trial	Grade 3–4 neutropenia in 73% of patients 32% required hospitalization 0.8% deaths related to toxicity	NR
Adjuvant Navelbine International Trialist Association (ANITA)	Grade 3–4 neutropenia toxicity in 92% of patients 9% febrile neutropenia 2% deaths related to toxicity	NR
MAGRIT trial	Grade ≥ 3 AE 2% of infections and infestations, 2% of vascular disorders, 2% of neoplasm	Grade ≥ 3 3% of infections and infestations 3% of vascular disorders 2% of neoplasm

Abbreviation: AE, adverse events; EGFR, epidermal growth factor receptor; TKI, tyrosine kinase inhibitor; NSCLC, non-small-cell lung cancer; NR, not reported.

**Table 4 cancers-13-04119-t004:** Past clinical trials that included adjuvant therapy using EGFR-TKI in patients with completely resected EGFR mutated * NSCLC.

Outcome: 2-Year Median DFS	TKI Duration	Control Arm	Treatment Arm	NSCLC Stage *	Participants (*n*)	Publication Date	Trial
78.9% (95% CI N/A) in AC with TKI vs. 54.2% (95% CI N/A) in AC without TKI (*p* value N/A)	6 months	Pemetrexed + carboplatin	Pemetrexed + carboplatin followed with gefitinib	IIIA	60	2014 March	Pemetrexed + carboplatin AC with or without gefitinib (Phase II)
81.4% (95% CI 69.6–93.1) in erlotinib arm vs 44.6% (95% CI 26.9–62.4) in chemotherapy arm (*p* = 0.0054)	2 years (median)	Vinorelbine + cisplatin	Erlotinib	IIIA	102	2018 August	EVAN (Phase II)
28.7 months with TKI (95% CI, 24.94–32.46) vs. 18 months with chemotherapy combination (95% CI, 13.59–22.34) (*p* = 0.005)	2 years	Vinorelbine + cisplatin	Gefitinib	II to IIIA	222	2017 November	ADJUVANT -CTONG1104 (Phase III)
88% in erlotinib arm (95% CI N/A)	2 years		Erlotinib (single arm, after AC)	IA to IIIA	100	2018 November	SELECT (Phase II)
89% (95% CI, 85–92) in osimertinib arm vs 52% (95% CI, 46–58) in placebo arm (*p* value N/A)	3 years	Placebo	Osimertinib	IB to IIIA	682	2020 September	ADAURA (Phase III)

Abbreviations NSCLC, non-small cell lung cancer; TKI, tyrosine kinase inhibitor; AC, adjuvant chemotherapy; HR, hazard ratio, DFS, disease-free survival, OS, overall survival. * EGFR mutations, Exon 19 deletion or exon 21 point mutation L858R, confirmed for all patients.

**Table 5 cancers-13-04119-t005:** Adverse Events reported for clinical trials that included adjuvant therapy using EGFR-TKI in patients with completely resected EGFR mutated NSCLC.

Adverse Events Control Arm	Adverse Events TKI Arm	Trial
NR	Approximately 43% of the patients who received both AC and gefitinib developed a rash	Pemetrexed-carboplatin adjuvant chemotherapy with/without gefitinib (Phase II)
Grade ≥ 3, in 11% of patients decreased neutrophil count 16% of patients. Myelosuppression: 9% of patients.	Grade ≥ 3, in 12% of patients Rash in 4% of patients.	EVAN (Phase II)
Grade ≥ 3 Neutropenia in 34% of patients Leucopenia in 16% of patients Vomiting in 8% of patients. Serious AE in 23% of patients	Grade ≥ 3: 2% of patients with elevated alanine aminotransferase and 2% of patients with elevated aspartate aminotransferase Serious AE in 7% of patients	ADJUVANT—CTONG1104 (Phase III)
Not relevant	No grade 4 or 5 AE. Grades 1–3A: rash, diarrhea, dry skin, fatigue, nausea/vomiting, nail changes, pruritis, stomatitis, and transaminitis. Recurrence occurred in 40% of patients 40% of patients required dose reduction of erlotinib, while 16% of patients required second dose reduction	SELECT (Phase II)
89% of patients reported AE Grade ≥3 AE reported in 13% of patients	Grade ≥3 AE reported in 20% of patients: diarrhea, paronychia, stomatitis, upper respiratory tract infection and decreased appetite. 98% of patients reported AE Interstitial lung disease in 3% of patients	ADAURA (Phase III)

Abbreviations AE, Adverse events, TKI, tyrosine kinase inhibitor NR, not reported.

**Table 6 cancers-13-04119-t006:** Ongoing clinical trials that include adjuvant therapy with EGFR-TKI in patients with completely resected EGFR mutated NSCLC.

Estimated Study Completion Date	Study Start Date	TKI Duration (years)	EGFR Mutation	Control Arms	TKI Arms	NSCLC Stage	Estimated Enrollment (*n*)	Trial
October 2026	August 2014	2	In exon 19 or L858R confirmed for all patients	Placebo	Erlotinib	IB to IIIA	450	ALCHEMIST [A081105] (Phase III)
December 2022	April 2011	1	In Exon 19 or 21	Gemcitabine/cisplatin (2 cycles) as neoadjuvant and adjuvant (2 cycles)	Erlotinib as neoadjuvant (42 days) and adjuvant (1 year)	IIIA	72	EMERGING (Phase II)
November 2021	January 2013	2	EGFR Mutations	Afatinib for 2 years	Afatinib for 3 months	I to III	92	Adjuvant Afatinib (Phase III)

Abbreviations: EGFR, epidermal growth factor receptor; TKI, tyrosine kinase inhibitor, NSCLC, non-small cell lung cancer.

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
