# Peer review of "Adjuvant Treatment with Tyrosine Kinase Inhibitors in Epidermal Growth Factor Receptor Mutated Non-Small-Cell Lung Carcinoma Patients, Past, Present and Future"

_cancers, 2021, doi:10.3390/cancers13164119_

Round 1

Reviewer 1 Report

Comments and suggestions

In this manuscript, entitled “Adjuvant Treatment with Tyrosine Kinase Inhibitors in Epidermal Growth Factor Receptor mutated Non-Small-Cell Lung Carcinoma Patients, Past, Present and Future”, authors review and summarize the clinical studies focusing on adjuvant therapy with TKIs in EGFR-mutated NSCLC. The authors introduced this topic in detail through a complete summary of clinical trial and research. Thus, I think this review provides useful and complete information on the clinical development of TKI in patients with EGFR-mutant NSCLC. In addition, the article is well organized and written fluently.

Reviewer 2 Report

Shalata et al. have conducted an interesting review based on the evaluation effect of adjuvant treatment with TKi in EGFR mutated NSCLC patients.  In an effort to further improve the strength of their manuscript, the Authors should take the following action.

Major revisions

The Authors  should modify and split all data presented in this review into two setting: 1) general population of NSCLC patients not characterized for EGFR (which may also included EGFRm patients but not identified in general population); and  2) population of selected patients  with EGFR mutated.

Minor revisions

1. Introduction

  1. Report the impact of stage on the magnitudo effect of the adjuvant treatment 
  2. Clarify the mutations of EGFR considered in the adjuvant clinical trials

2. Past clinical trials that included adjuvant chemotherapy (AC) without EGFR TKi

  1. Table 2: Specify the outcomes for each reported study
  2. Table 3: Report the Adverse events in a homogeneous and comparable method for all clinical trials

3. EGFR TKi user in clinical trials

  1. 4.1 Pemtrexed.carboplatin adjuvant chemotherapy with/without gefitinib Trial. Check PFS is not correct, you should replace with DFS
  2. Table 4: Same replace PFS with DFS
  3. Table 5: Repot the Adverse events in a homogeneous and comparable method for all clinical trials

4. Discussion

  1. Report the knowledge on the natural history and different biology of NSCLC with and without EGFR mutations
  2. Report the data on  impact of the stage in the adjuvant treatment
  3. Clarify the difference between immunocheckpoint inhibitors and TKi in adjuvant treatment of NSCLC patients
  4. Report the significance of different mutations in adjuvant treatment of EGFR mutated NSCLC

5 Conclusions

  1. Report the prospective in peri-operative and adjuvant treatment in EGFR mutated NSCLC patients

Reviewer 3 Report

Dear Authors,

The article "Adjuvant Treatment with Tyrosine Kinase Inhibitors in Epidermal Growth Factor Receptor Mutated Non-Small-Cell Lung Carcinoma Patients, Past, Present and Future" written by W.  Shalata et al, is focused on highlighting therapeutic strategies in a cohort of patients with non-small cell Lung Cancer (NSCLC) carrying mutation in  epidermal growth factor receptor (EGFR). Therefore the current review aimed to summarize and characterize the data from past clinical trials based primarily on specific chemotherapy (tyrosine kinase inhibitors (TKIs) as well as recent advances in application of this type of therapy.

All the suggestions below reflect my personal view and are aimed to be taken for consideration. 

  1. I would suggest to highlight in Title that review is limited by trials with resectable stages of NSCLC.
  2. In my personal opinion, it is difficult to find logic in the way the information is presented. I assume that the authors want to present the data in a chronological manner. For that reason I suggest to highlight the story in introduction and tell that adjuvant therapy after resection is dramatically changed that it was before TKIs were introduced as supportive and later as a monotherapy for patients with NSCLC. 
  3. Introduction is very informative and compact. It is nice to have a definition of NSCLC, its types, histological variants, EGFR,  and classical treatment strategies and survival rate. But I would suggest moving the information about generations of TKI to chapter 3. Instead I would suggest giving brief information regarding stages, to remind readers of the difference between for example resectable and non-resectable stages (I-IIIA and IIIB/IV). Because only I-IIIA are the focus of the current study.
  4. I'm  wondering why information about forming resistance is missing. I would suggest adding a paragraph depicting concerns and contraindications for TKI and the resistance that forms as a limiting factor in treatment of NSCLCs.
  5. Future perspective is missing. It would be nice if the authors will summarize data and reveal key directions in treatment strategies and discuss more about future perspectives and link to immunotherapy. 

Unfortunately the downloaded version does not contain line numbers. 

Here I add some minor suggestions. 

Line:

In the current review, we have analyzed the data from clinical trials in the past and and highlighted the progress made in treatment of non-small-cell lung carcinoma (NSCLC), with a special focus on adjuvant therapy using tyrosine kinase inhibitors (TKIs) administered to NSCLC patients who had a complete resection of the tumor harboring a mutated epidermal growth factor receptor.

Line: Most reviewers concluded that the combination of cisplatin plus vinorelbine is probably the best choice for adjuvant treatment[8,9]. Change to:  The most recent findings show that …. the combination of cisplatin plus vinorelbine is probably the best choice for adjuvant treatment.

Line: There are five TKI currently available for treatment of NSCLC in patients with EGFR-mutations, which are divided into 3 generations.

Line: In this review paper we analyzed and summarized recent findings in treatment and efficacy of TKI for NSCLC patients with EGFR mutations.

Round 2

Reviewer 2 Report

The draft cannot be accepted even in this revision as it is confusing.  Studies identified as “past Chemtherapia without TKi” should be considered separately both in the text and in all tables.  This is because they refer to patient populations enrolled in studies in which EGFR status had not historically been assessed.  Therefore the data from these studies cannot be compared with those from studies in patient populations characterized for EGFR mutations

Reviewer 3 Report

Dear Authors,

The majority of my concerns and comments were addressed.

Author Response

it was our pleasure for your comments

Round 3

Reviewer 2 Report

It is necessary both in the text and in the tables to clearly separate the studies considered in: 1) populations uncharacterized for the state of EGFR ("past") and 2) mutated EGFR populations. Therefore the term "chemotherapy / adjuvant therapy without EGFR-TKi" should be replaced with the term "uncharacterized for EGFR"

Round 4

Reviewer 2 Report

The authors should also modify (as already done in table 1) the titles of paragraph 2, table 2, table 3 and text using more correctly the phrase "chemotherapy in trials uncharacterized for EGFR status" instead of "therapy without EGFR-TKI"